# Inherited Retinal Degeneration Caused by Dehydrodolichyl Diphosphate Synthase Mutation–Effect of an *ALG6* Modifier Variant

**DOI:** 10.3390/ijms25021004

**Published:** 2024-01-13

**Authors:** Elisha Monson, Artur V. Cideciyan, Alejandro J. Roman, Alexander Sumaroka, Malgorzata Swider, Vivian Wu, Iryna Viarbitskaya, Samuel G. Jacobson, Steven J. Fliesler, Steven J. Pittler

**Affiliations:** 1Department of Optometry and Vision Science, University of Alabama at Birmingham, Birmingham, AL 35294, USA; eli2000@uab.edu; 2Center for Hereditary Retinal Degenerations, Scheie Eye Institute, Department of Ophthalmology, Perelman School of Medicine, University of Pennsylvania, Philadelphia, PA 19104, USA; aroman@pennmedicine.upenn.edu (A.J.R.); asumarok@pennmedicine.upenn.edu (A.S.); mswider@pennmedicine.upenn.edu (M.S.); vivian.wu@pennmedicine.upenn.edu (V.W.); iryna.viarbitskaya@pennmedicine.upenn.edu (I.V.);; 3Departments of Ophthalmology and Biochemistry, and Neuroscience Graduate Program, Jacobs School of Medicine and Biomedical Sciences, State University of New York—University at Buffalo, Buffalo, NY 14203, USA; fliesler@buffalo.edu; 4Research Service, VA Western NY Healthcare System, Buffalo, NY 14215, USA; 5Vision Science Research Center, School of Optometry, University of Alabama at Birmingham, Birmingham, AL 35294, USA

**Keywords:** genetic modifier, retinitis pigmentosa, *DHDDS*, glycosylation

## Abstract

Modern advances in disease genetics have uncovered numerous modifier genes that play a role in the severity of disease expression. One such class of genetic conditions is known as inherited retinal degenerations (IRDs), a collection of retinal degenerative disorders caused by mutations in over 300 genes. A single missense mutation (K42E) in the gene encoding the enzyme dehydrodolichyl diphosphate synthase (DHDDS), which is required for protein N-glycosylation in all cells and tissues, causes *DHDDS*-IRD (retinitis pigmentosa type 59 (RP59; OMIM #613861)). Apart from a retinal phenotype, however, *DHDDS*-IRD is surprisingly non-syndromic (i.e., without any systemic manifestations). To explore disease pathology, we selected five glycosylation-related genes for analysis that are suggested to have disease modifier variants. These genes encode glycosyltransferases (*ALG6*, *ALG8*), an ER resident protein (*DDOST*), a high-mannose oligosaccharyl transferase (*MPDU1*), and a protein N-glycosylation regulatory protein (*TNKS*). DNA samples from 11 confirmed *DHDDS* (K42E)-IRD patients were sequenced at the site of each candidate genetic modifier. Quantitative measures of retinal structure and function were performed across five decades of life by evaluating foveal photoreceptor thickness, visual acuity, foveal sensitivity, macular and extramacular rod sensitivity, and kinetic visual field extent. The *ALG6* variant, (F304S), was correlated with greater macular cone disease severity and less peripheral rod disease severity. Thus, modifier gene polymorphisms may account for a significant portion of phenotypic variation observed in human genetic disease. However, the consequences of the polymorphisms may be counterintuitively complex in terms of rod and cone populations affected in different regions of the retina.

## 1. Introduction

Glycoproteins are an important class of biomolecules that are ubiquitously involved in many biological processes, including cell–cell recognition, immune response, extracellular matrix formation, ion and solute transport, and signal transduction [1,2]. The transfer of glycans to proteins involves a complex pathway that is well-studied but not yet fully understood [3,4,5]. Mutations in genes encoding components of this pathway can result in defective glycosylation and, hence, one or more congenital disorders of glycosylation (CDGs) [6,7,8,9]. CDGs typically manifest most profoundly in the brain, liver, and kidneys, although most bodily tissues are affected in some way [6,7,8]. Common symptoms include, but are not limited to, ataxia, epilepsy, hypotonia, and visual impairment [7,8].

The anabolism and final transfer of glycans to nascent proteins to form glycoproteins are mediated by dolichol, a transmembrane isoprenoid lipid [10,11,12]. Dolichol synthesis and subsequent protein N-glycosylation is dependent on *cis*-prenyltransferase (cPT), which is a heterodimer of four subunits (two subunits encoded by dehydrodolichyl diphosphate synthase *(DHDDS)* and two by Nogo-B receptor *(NgBR)*). Counterintuitively, a founder mutation (c.124A→G, p.K42E) in *DHDDS* causes only an autosomal recessive inherited retinal degeneration (IRD) known as RP59 [13,14,15]. Despite the ubiquitous requirement for glycosylation in all mammals, *DHDDS*-IRD is non-syndromic, placing it in the retinitis pigmentosa disease category with ~90 other genes (RetNet, https://web.sph.uth.edu/RetNet/sum-dis.htm?csrt=18325746694721182734#A-genes; accessed on 5 December 2023) [16]. While IRDs in general have an estimated prevalence of 1:1000–1:5000, in the Ashkenazi Jewish population, the K42E mutation causing *DHDDS*-IRD has a much higher prevalence, estimated to be 1:330 [15,17]. RP59 (OMIM #613861) exhibits classic features of RP including bone spicule-like pigmentation, waxy pallor of the optic disc, and attenuated retinal vessels, all apparent on fundus examination. Additionally, retinal degeneration particularly of the photoreceptors is observed using OCT and the ERG is diminished due to compromised photoreceptor function. The rate of progression is variable, with onset in the early 20s and progressive degeneration.

Characterization of monogenic inherited conditions in humans is severely complicated by many confounding factors, including variable phenotypic expression, allelic heterogeneity, and/or incomplete penetrance. Unrelated individuals harboring the same causative mutation but deviating from each other in genetic background can exhibit phenotypic differences ranging from mild variation to complete absence of the disease state [18,19,20,21]. Genetic conditions implicated with such confounding factors represent a wide range of human diseases [22,23,24]. Rahit and Tarailo-Graovac (2020) [19] defined genetic modifiers as “genetic variants that can modify the phenotypic outcome of a primary disease-causing variant” without necessarily being pathogenic alone. Recently, the topic of modifier genes has gained interest due to their application in clinical diagnostics and insight into disease mechanisms. The means by which these factors affect the phenotype vary greatly between modifiers [19].

*ALG6* F304S manifests classic indications of a modifier gene variant: asymptomatic occurrence in healthy populations and modulated disease severity of a primary disease. For example, light-induced retinal degeneration (LRD) is a condition initiated by prolonged exposure of the retina to bright light. Puzzled by significant phenotypic differences between murine breeds, researchers performed genome analyses on a group of similar mouse models exhibiting a diverse range of LRD phenotypes [25]. Their findings exposed a difference between mouse strains in amino acid 450 of murine retinoid isomerohydrolase (RPE65) from leucine to methionine, with the Met450 strain experiencing less degeneration from LRD [25]. This variant was linked to reduced levels of RPE65 and the resulting decrease in rhodopsin regeneration and light absorption [26]. Mice with the L459M variant were better able to endure excessive light exposure and retain photoreceptor cells/function [26]. Our goal in studying this potential *ALG6* modifier is to likewise understand its mechanism of effect on *DHDDS*-IRD (RP59).

IRDs are a class of diseases that exhibit significant, yet unexplained, clinical variation and, as such, are a prime target for modifier gene analysis [27]. Studies in this area have yielded many useful insights into genetic disease modification. For example, the gene retinitis pigmentosa GTPase regulator (*RPGR*) is found on the human X-chromosome and encodes a protein associated with the so-called “connecting cilium” of retinal rod photoreceptor cells. Mutations in this gene are the most common cause of X-linked retinitis pigmentosa (XLRP), a severe form of RP [28]. *CEP290*, a widespread ciliopathy-associated gene, can present with a homozygous *rd16* mutation that leads to complete photoreceptor degeneration. However, when present in the hypomorphic heterozygous state in male *RPGR* knockout mice, *Cep290*^rd16/+^ genotypes exhibit a much earlier onset of RPGR-XLRP [28]. In addition, many other severity-altering variants have also been reported for IRDs [29,30,31,32,33].

With its non-syndromic and yet-to-be-fully elucidated disease mechanism, *DHDDS*-IRD is an excellent target for modifier gene analysis. Additionally, the homogeneity resulting from all patients carrying the same founder mutation eliminates the difficulties associated with allelic heterogeneity. In this study, five potential modifier variants were chosen, based on a clear association with protein N-glycosylation, but no inherent pathogenicity. Each variant was reported from whole exome sequencing data of an infant male presenting with a significant reduction in *DHDDS* mRNA (residual 20% of normal control levels) and reduced enzymatic activity (residual 35% of normal control levels) leading to subsequent death at eight months of age [34]. Four of these potential modifiers were not previously reported, while the fifth (*ALG6*, F304S) was suggested to alter *ALG6* functionality and potentially correlate with CDG severity in several CDG-Ia-causing variants [35,36]. Here, we report on the analysis of clinical correlations of five candidate variants with disease severity in eleven cases of *DHDDS*-IRD.

## 2. Results

### 2.1. Selection of Potential Modifier Variants

Five candidate modifier variants (Table 1) were selected for genetic analysis based on their relevance to glycosylation pathways and lack of previous identification with related pathologies. Each of these putative modifiers was initially identified in a *DHDDS* patient clinical report [34]. *ALG6* (encoding dolichyldiphosphate Man9GlcNAc2 α1,3-glucosyltransferase) and *ALG8* (encoding dolichyldiphosphate Glc1Man9GlcNAc2 α1,3-glucosyltransferase) are genes coding for glycosyltransferases directly involved in glucose additions to dolichyl phosphate-linked oligosaccharides within the lumen of the ER [37]. *DDOST* codes for an enzyme (dolichyl-diphosphooligosacccharide-protein glycosyltransferase) required for the transfer of dolichol-linked high-mannose type oligosaccharides to asparagine residues in the N-glycosylation consensus sequence (Asn-X-Ser/Thr) of nascent acceptor proteins in the lumen of the ER [38]. *MPDU1* encodes mannose-P-dolichol utilization defect 1, an ER resident enzyme required for the synthesis of mannose-P-dolichol, an essential substrate for protein N-glycosylation [39,40]. *TNKS* (Tankyrase) has many functions within a cell, one of which is the regulation of glycosylation [41]. Two of the chosen variants were synonymous single nucleotide polymorphism (sSNP) variants, which, despite leaving amino acid sequences unchanged, can have many deleterious effects. For example, sSNPs have been linked to defects in mRNA secondary structure, protein folding, and protein function [42,43,44].

### 2.2. DNA Sequencing Results Shows That Only ALG6 Displays Sequence Variation

To assess the presence or absence of identified potential phenotypic modifiers (Table 1), we PCR-amplified patient DNA with primer sets that spanned each potential modifier variant to determine the DNA sequence.

All DNA sequence data were viewed as chromatograms (Figure 1) and checked for acceptable background levels before being analyzed (Table 2). While some minor background was apparent in some samples (e.g., Figure 1A,E), all DNA sequencing runs yielded clear results that were easily readable without ambiguity. Collected sequences showed that no change from the most common sequence in the population was observed for four of the five potential variants analyzed in *ALG8*, *DDOST*, *MPDU1*, and *TNKS*. *ALG6* sequence analysis did show variation, however, with two control individuals (one homozygous and one heterozygous) and five patients (all heterozygous) expressing the variant while the remaining six patients showed a normal allele sequence.

### 2.3. Differences in Severity of Disease in Patients

The form of autosomal recessive IRD caused by mutations in *DHDDS* can exhibit different patterns of disease distribution across the retina [13,14,46,47,48,49,50] and this is exemplified by the detailed imaging and functional studies performed with two patients from the current cohort (Figure 2). P1 (Table 3) reported the onset of nyctalopia at age 18. At age 24, he had best corrected visual acuity (BCVA) values of 20/16 and 20/20 for the right and left eyes, respectively. Centrally, en face autofluorescence images showed a spatial distribution of disease that is typical for RP with a retained region encompassing the fovea and parafovea surrounded by more extensive pathology at greater eccentricities (Figure 2A, left). There was an atypical parapapillary preservation that is seen in some other IRDs [51,52,53] but previously not reported for *DHDDS*. Cross-sectional imaging with optical coherence tomography (OCT) showed a retained foveal outer nuclear layer (ONL) surrounded by thinning ONL consistent with greater retinal degeneration (Figure 2B, blue highlight). Rod-mediated sensitivity with dark-adapted 500 nm stimuli (DA 500) was severely reduced everywhere except nasal to the fovea, including the parapapillary region (Figure 2C, upper). Cone-mediated sensitivity with light-adapted 600 nm stimuli (LA 600) was near-normal centrally and in the parapapillary region but reduced with greater eccentricity (Figure 2C, lower). Kinetic visual fields were symmetric and showed mild constriction with the V-4e target to 60–70° diameter and with the I-4e target to ~30° diameter.

P5 (Table 3) reported the onset of nyctalopia in her early 20s. At age 31, she had BCVA values reduced to 20/60 and 20/70 for her right and left eyes, respectively. En face imaging of RPE disease with NIR-RAFI and SW-RAFI demonstrated a severe macular disturbance (Figure 2A, right) that transitioned to healthy retina near the eccentricity of the optic nerve. OCT showed partially retained foveal ONL surrounded by severe degeneration and further surrounded by near normal ONL thickness in the extramacular region (Figure 2B, right). Rod-mediated sensitivity was severely abnormal centrally but reached near-normal levels in the extramacular retina (Figure 2C, upper). Cone-mediated sensitivities showed a pericentral reduction (Figure 2C, lower). Kinetic visual fields were symmetrical and nearly full to the V-4e target but showed severe constriction to a foveal tunnel with the I-4e target.

Both patients were clinically diagnosed as RP (Table 3) but showed different patterns of macular and extramacular disease that could not be sequenced into a single underlying natural history demonstrating the difficulty of defining retinal disease severity with simpler measures. In the eleven patients analyzed here, clinical diagnoses of RP (in most patients) versus cone-rod degeneration (CRD) (in one family) did not correlate with the *ALG6* variant, and limited ERG findings were not informative. Specifically, 10 of 11 patients had recordings performed at ages varying from 17 to 56 years and both rod and cone ERGs were either not detectable or severely attenuated. P11 at age 13 had a nondetectable cone ERG and a reduced but detectable rod ERG.

To consider the effects of the *ALG6* variant on quantitative measures of visual function and structure, we distinguished four parameters derived from fovea or macula (Figure 3A–D), one parameter that could represent fovea, macula, or extra-macular function depending on the stage of disease (Figure 3E), and one parameter that originated from the extra-macular region (Figure 3F). One eye was chosen per patient for all analyses. Loss of visual acuity tended to start sooner in patients with the *ALG6* variant (Figure 3A, colored symbols), but this effect was statistically not significant (*p* = 0.077). Foveal cone sensitivity loss also tended to appear sooner in patients with the *ALG6* variant (Figure 3B), but the difference between groups was not statistically significant (*p* = 0.35). Progressive thinning of the foveal ONL showed a tendency for greater severity in patients with the *ALG6* variant (Figure 3C), but this did not reach statistical significance (*p* = 0.39). Lastly, the time course for loss of peak rod sensitivity within the macula (Figure 3D) did not show a difference between groups (*p* = 0.86). The extent of the kinetic visual field is a complex measure mediated by cone photoreceptors located in the macular or extramacular retina depending on the stage of disease. Progressive constriction of the kinetic visual field (Figure 3E, colored symbols) showed no obvious difference between groups with or without the *ALG6* variant (*p* = 0.35). Last considered was the rod photoreceptor-mediated sensitivity in the extramacular region, which showed a very large spectrum of results from near-normal values to losses of nearly five log units. Unexpectedly, patients with the *ALG6* variant showed substantially less severity, retaining peak rod function 14.2 years longer (Figure 3F) on average, and the difference between groups was statistically significant (*p* = 0.018).

Our results in 11 patients were generally consistent with a large cohort of other patients with *DHDDS* mutations previously published [50] in terms of the natural history of disease implied from BCVA (Figure 3A, gray symbols) and kinetic visual field constriction (Figure 3E, gray symbols). Thus, the six variables of disease severity considered in the current study led to the hypothesis that the *ALG6* variant in *DHDDS* patients may be associated with incrementally earlier degeneration of macular cones but substantially later degeneration of the extra-macular (peripheral) rods.

## 3. Discussion

Modifier genes have been implicated in altering the severity of disease phenotype in a number of hereditary disorders, including CDGs and IRDs [19,27,54]. Thus, it was worthwhile to assess several potential modifier alleles for implicated variants in *DHDDS*-IRD. All eleven patients analyzed in this study were successfully verified to carry the primary disease-causing mutation, K42E, in contrast to three control individuals exhibiting the reference sequence (K42). Of the five potential modifier variants analyzed, only one (*ALG6* F304S) expressed genotypic variation. Five individuals presented as heterozygous for the *ALG6* variant while the remaining six patients were homozygous for the reference sequence. To further support the previous classification as a non-pathogenic variant, two out of three controls also expressed this alteration [55].

Of the six clinical parameters used to determine retinal function and structure, four (visual acuity, foveal cone sensitivity loss, foveal ONL thickness, and macular rod sensitivity loss) are indicative of only macular health, not overall retinal health. No change was observed in macular rod sensitivity loss; however, a trend was observed in the other three parameters collectively suggesting diminished macular cone photoreceptor health in individuals heterozygous for the *ALG6* variant. A fifth parameter, extra-macular visual field extent, which is informative with regard to cone photoreceptor function in the periphery, indicated no appreciable change. A sixth parameter analyzed (extra-macular rod sensitivity loss), however, showed significantly delayed peripheral rod degeneration in patients heterozygous for the *ALG6* variant (Figure 3F; *p* = 0.018). Overall, these results indicate a potential deficit in macular cone function and simultaneous preservation of peripheral rod health in *DHDDS*-IRD patients co-expressing a heterozygous F304S mutation in *ALG6*.

The *ALG6* F304S variant was first reported in 2000 where it was found homozygously with another missense variant in two novel clinical cases of *ALG6*-CDG (CDG-Ic) [35]. A modified strain of *S. cerevisiae* was subsequently used to test ALG6 with the F304S change for any adverse effect on protein glycosylation. While glycosylation in the modified yeast cells was comparable to WT strains, a severe impairment of ALG6 function was also observed. Later that year another clinical case was published, describing a six-year-old male presenting with classic symptoms of a CDG [45]. Both parents (who were asymptomatic) contributed *ALG6* variants to their son, including a heterozygous F304S variant and in-frame deletion (D299) from the father. Expression of F304S-modified *ALG6* in yeast cells, however, did not result in a detectable reduction in normal protein glycosylation. However, the D299/F340S paternal allele in the patient was noticeably underrepresented, prompting the hypothesis that either/both of these variants could result in transcriptional defects or RNA instability. Vuillaumier-Barrot et al. (2001) published a report showing that *ALG6* F304S had an allele frequency of 27% in the French population [55]. This prompted the conclusion that F304S is a non-pathogenic variant, a finding supported by a similar Croatian study [56]. The real breakthrough, however, indicating a modifier effect came in a large-scale patient study exploring the prevalence of F304S in *PMM2*-CDG (CDG-Ia) patients [36]. In that study, the F304S variant occurred at significantly higher rates (0.41 vs. 0.21 and 0.36 vs. 0.18) in patients with severe and fatal cases, respectively, of CDG-Ia. More recently, a primary *DHDDS* case study describing a male infant who also was homozygous for the *ALG6* F304S variant and succumbed to severe CDG-I at eight months of age [34] led the authors to speculate that the ALG6 variant acted as a phenotypic modifier. These findings are consistent with the ALG6 F304S variant increasing the severity of primary causal variants in other genes.

To consider the mechanism of F304S modifier activity on *DHDDS*, it is necessary to understand glycosylation pathways. The protein N-glycosylation and other protein glycosylation pathways require more than 35 enzymes and are crucial for function in all cells of the body [57,58,59,60,61]. During N-glycosylation, GlcNAc_2_Man_9_Glc_3_ carbohydrate complexes are gradually assembled on an ER transmembrane lipid carrier (dolichol-P) and finally transferred to specific asparagine residues on target proteins. The enzyme *ALG6* adds the first of the final three glucose residues to the pre-assembled GlcNAc_2_Man_9_ sugar [37]. Thus, knowledge of the structure of this enzyme may provide insight into the potential modifier effect of F304S. *ALG6* is composed of 507 amino acids and is predicted to contain 11 transmembrane helices that span the ER membrane Nextprot, https://www.nextprot.org/entry/NX_Q9Y672/structures (accessed on 15 March 2023) [62,63]. The *ALG6* active site has yet to be clearly delineated in humans, but in yeast analogs, it was shown to use aspartic acid 69 as part of its catalytic site within the ER lumen [37]. Despite over twenty different mutations/variants in *ALG6* being linked to *ALG6*-CDG (CDG-Ic), none of them overlap with the corresponding active site in yeast [37]; in fact, most deleterious mutations occur within the transmembrane regions of the enzyme where dolichol carriers of glucose likely bind [62,64]. This highlights the crucial role that these regions have in *ALG6* structure, function, and/or anchoring. Not surprisingly, F304S is also located deep within the ER membrane on transmembrane helix six, far from the predicted active site.

In contrast to *ALG6* function, the cPT enzyme (DHDDS + NgBR) is not directly involved in the assembly of glycans, normally only acting to generate dolichol, the obligate glycan carrier. This functional and physical separation (*ALG6* is in the lumen of the ER and cPT is cytosolic) makes the mechanistic basis for the potential modifier effect of *ALG6* on *DHDDS* more difficult to understand. All of the enzymes involved in glycosylation are, however, tightly compartmentalized within the ER and could cross paths during transport from the site of synthesis to their sites of action. Normally, these enzymes would pass without interaction, but the conformational changes (*ALG6* F304S and *DHDDS* K42E) could lead to increased favorability of interaction, diminishing full functional capacity and thereby exacerbating the disease state—a deleterious gain of function. Applying this to the observed preservation of extra-macular rod sensitivity loss in our *DHDDS* patients suggests a more prominent effect in macular cones and a previously unknown difference in glycosylation metabolism in macular cones vs. extra-macular rods. Further studies with more patients will be necessary to determine the prevalence of *ALG6* modifier activity on *DHDDS* disease and to consider interventions that could slow vision loss [65,66].

In summary, this study emphasizes the importance of a broader view of genome involvement, even in single-gene disorders. A major basis for phenotypic heterogeneity in hereditary degenerative disease is likely modifier SNPs in other genes. Many modifier variants that influence disease phenotype have been reported for a wide array of hereditary diseases [18,19,20,21,22,23,24,25,26,27,28,29,30,31,32,33,34,35,36,45,54,55,56,65,66]. To understand the broad spectrum of genetic disease, including but well beyond ocular disease, a systematic means of global consideration of genome sequence using WES or WGS will be needed, which is now possible due to advances in genome technology.

## 4. Materials and Methods

### 4.1. Patient DNA Procurement

DNA samples were purified from peripheral blood in three healthy reference individuals and eleven patients (eight unrelated patients and three siblings from one family) who were known to harbor homozygous K42E mutations in the gene *DHDDS*. To ensure “masked” (agnostic) experimentation, clinical severity for each patient was kept undisclosed until all sequencing analysis was finalized.

### 4.2. Clinical Assessments

Patients underwent a standard ophthalmic examination including best corrected visual acuity (BCVA) and Goldmann kinetic perimetry with a V-4e target. The extent of the kinetic visual field was quantified by using a computer-based algorithm and presented as a percent of the mean normal visual field extent [67,68]. Electroretinogram (ERG) recordings were performed or reported in a subset of ten patients. Specialized testing was performed when available and appropriate. Static perimetry was performed with a modified automated perimeter (Humphrey Field Analyzer 750i, Carl Zeiss Meditec, Dublin, USA) as previously described [69,70,71]. In brief, orange (600 nm) stimuli were used under light-adapted conditions, and blue (500 nm) and red (650 nm) stimuli were used under dark-adapted conditions. Test patterns sampled the retina at 2° intervals across the central visual field (central 60° along horizontal and vertical meridians) and at 12° intervals throughout the visual field. Photoreceptor mediation under dark-adapted conditions was determined by the sensitivity difference between the two stimuli. Optical coherence tomography (OCT) was performed with either time-domain (Zeiss Humphrey Instruments, Dublin, CA, USA) or spectral-domain (RTVue-100; Optovue Inc., Fremont, CA, USA) instruments as previously described [71,72,73]. En face images were obtained using a confocal scanning laser ophthalmoscope (SLO; Spectralis HRA, Heidelberg Engineering, Heidelberg, Germany) to determine retinal pigment epithelium (RPE) integrity. Near-infrared reduced-illuminance autofluorescence images (NIR–RAFI) and short-wavelength reduced-illuminance autofluorescence imaging (SW-RAFI) were acquired as previously described [71,74].

### 4.3. PCR Amplification and DNA Sequencing

PCR primers (Thermo Fisher Scientific, Waltham, MA, USA) were designed using the UCSC Genome Browser’s GRCh37/hg19 and GRCh38/hg38 human assemblies and NCBI Primer Blast software (https://www.ncbi.nlm.nih.gov/tools/primer-blast/, accessed on 11 December 2023) to amplify ~200–600 base pairs surrounding each modifier variant and the *DHDDS* K42E mutation for verification (Table 4) [75]. PCR analysis was performed using GoTaq^®^ Master Mix (Promega, Tokyo, Japan, Cat. #M7132) in BioRad iCycler™ or MyCycler™ PCR machines. Methods for PCR analysis were essentially as described in detail previously [76]. Prior to DNA sequencing, all PCR products were gel-purified using a GeneJET Gel Extraction^®^ kit (Thermo Fisher Scientific, Cat. #K0831), diluted to ~6–40 ng/μL for DNA sequencing at the UAB Heflin Center for Genomic Sciences. One *ALG6* sample required a co-solvent (SequenceR_x_ Enhancer Solution A^®^, Thermo Fisher Scientific, Cat. #12238010) to improve sequence readability.

A discrepancy was detected in the literature with the *DDOST* variant used in our study. It was reported as c.679A→G, p.I227V [34]; however, according to the referenced genome database in Sabry et al. (2016) (UCSC genome browser, hg19 reference sequence), c.679A begins codon 227, which is AAC encoding asparagine (N), not isoleucine (I). Upon inquiry, an author of the Sabry et al. (2016) study informed us that their observations actually indicated a change in codon 244 (c.730A→G, p.I244V, personal communication, Dr. Sandrine Vuillaumier). Therefore, in accordance with the database sequence, we suggest that the correct nucleotide at position c.679 is indeed G while the most common sequence at c.730 is A and the mutated sequence is c.730G. A more recent study indicates that this sequence change occurs benignly in 0.02% of African persons [77]. Thus, we conclude that neither a c.679, nor a c.730 sequence change can act as a disease phenotype modifier.

### 4.4. Statistical Analyses

Linear mixed-effects models were used to assess progression with six variables consisting of visual acuity, foveal sensitivity, foveal ONL thickness, macular peak rod function, kinetic visual field extent, and extra-macular peak rod sensitivity. Separate regressions of each variable vs. Age and Group (WT/HET) were performed using a mixed-effects model accounting for the correlation structure of the data. The p-values given for differences between groups correspond to the Group term in each regression.

## Figures and Tables

**Figure 1 ijms-25-01004-f001:**
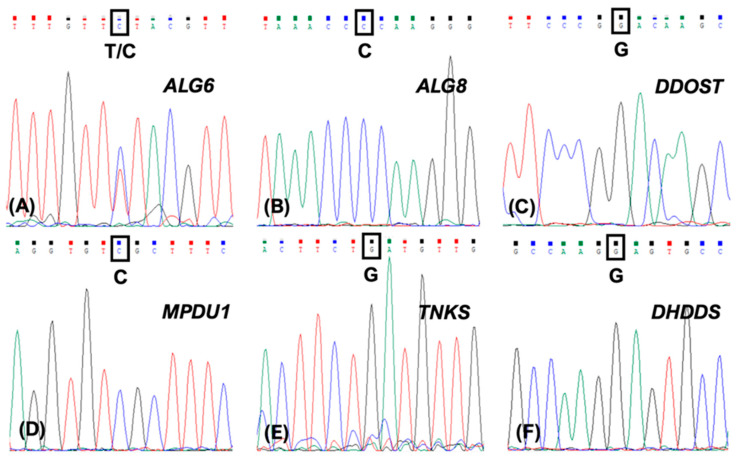
Representative chromatograms for each potential modifier gene. DNA from each patient sample was PCR-amplified at all five modifier sites and purified prior to DNA sequence analysis. Single nucleotide changes are marked with a black rectangle and enlarged for visibility below each rectangle. (**A**) *ALG6* (**B**) *ALG8* (**C**) *DDOST* (**D**) *MPDU1* (**E**) *TNKS* (**F**) *DHDDS*. Only the T→C variant in *ALG6* was found in some patient samples.

**Figure 2 ijms-25-01004-f002:**
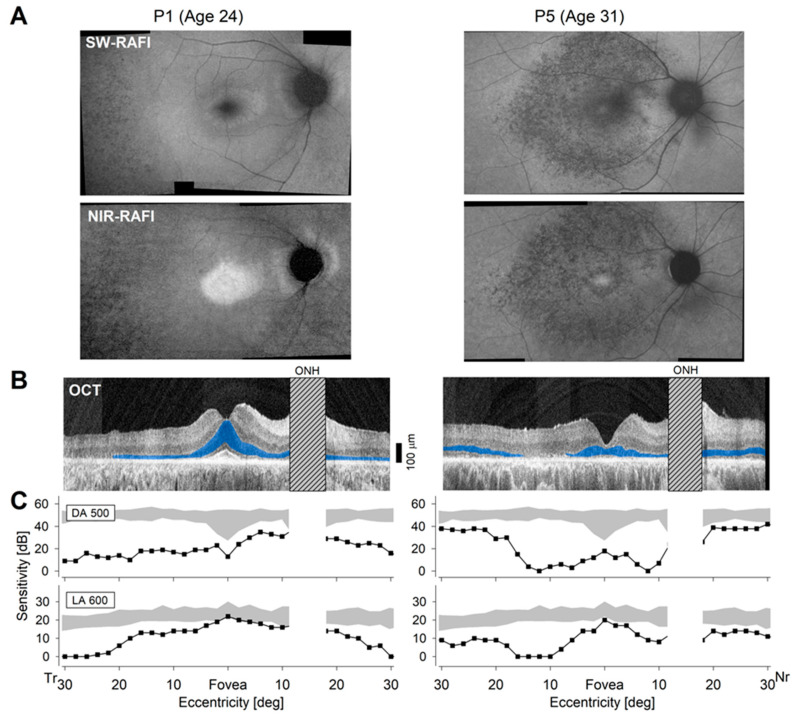
Phenotypes of two *DHDDS* patients showing two distinct distributions of retinal disease. (**A**) En face reduced-illuminance autofluorescence imaging (RAFI) with short-wavelength (SW) or near-infrared (NIR) excitation to evaluate disease distribution based on lipofuscin- or melanin-related pigments, respectively, across the RPE. (**B**) Cross-section imaging with optical coherence tomography (OCT) along the horizontal meridian crossing the fovea. The outer nuclear layer (ONL) is highlighted blue and the optic nerve head is hashed. (**C**) Sensitivity profiles sampled across the horizontal meridian crossing fixation using dark-adapted 500 nm (DA 500) stimuli mediated by rods and light-adapted 600 nm (LA 600) stimuli mediated by cones. Gray regions are normal limits and physiological blind spot is whited out.

**Figure 3 ijms-25-01004-f003:**
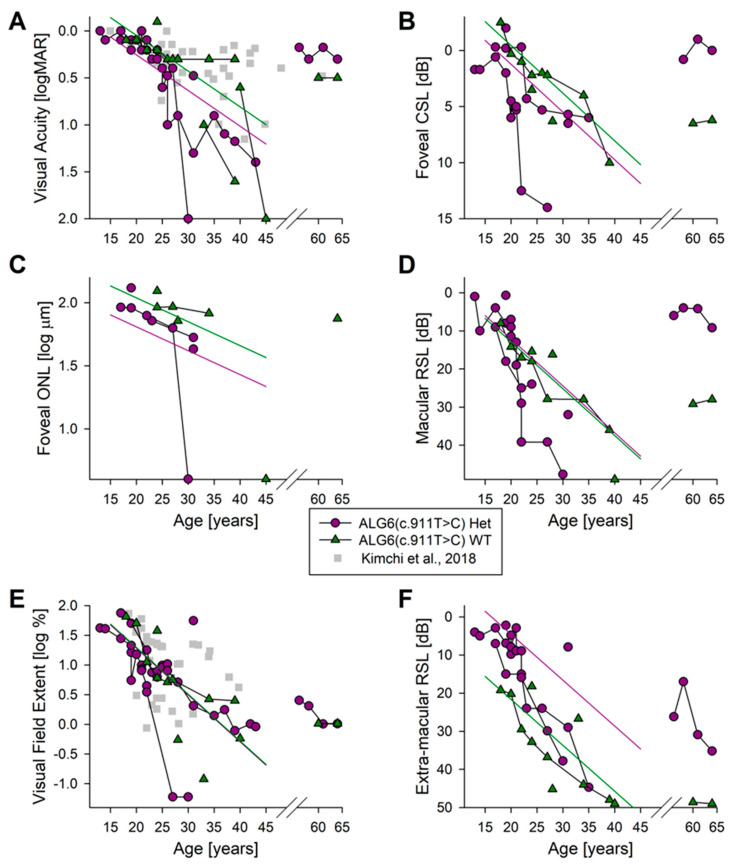
*ALG6* variant and retinal disease severity in *DHDDS* patients. (**A**) Best corrected visual acuity. (**B**) Foveal cone sensitivity loss (CSL). (**C**) Foveal outer nuclear layer (ONL) thickness. (**D**) Least rod-mediated sensitivity loss (RSL) representing best rod function within the confines of the macula. (**E**) The extent of the kinetic visual field (as a percent of mean normal) to a Goldman V-4e test target. All data originating from the macula are censored such that results represent the extramacular visual function. (**F**) Peak rod-mediated sensitivity (least RSL) representing best rod function outside the macula. In all panels, patients with *ALG6* wildtype are shown with green up triangles, and *ALG6* variant are shown with pink circles. Serial data obtained from the same eye are connected by lines. In panels (**A**,**E**), results reported in the literature [50] are digitized and plotted (gray squares) for reference. Kinetic visual field extent published in the literature uses a different target (III-4e) which was normalized using 183 cm^2^ as the mean normal value to plot on the same ordinate.

**Table 1 ijms-25-01004-t001:** Exonic potential modifier variants analyzed in this study.

Gene	Location	Protein	Mutation Type	Reference
*ALG6*	c.911 T→C	F304S	Missense	[34,35,45]
*ALG8*	c.1068C→G	P356 = **	Point (synonymous)	[34]
*DDOST*	c.679A→G	I227V	Missense	[34]
*MPDU1*	c.393C→T	V131 = **	Point (synonymous)	[34]
*TNKS*	c.1945G→A	D649N	Missense	[34]

** (=) indicates a synonymous change that does not alter the amino acid sequence.

**Table 2 ijms-25-01004-t002:** DNA sequencing results of all five variants and the *DHDDS* variant.

Patient	*ALG6*	*ALG8*	*DDOST*	*MPDU1*	*TNKS*	*DHDDS*
P1	−/−	−/−	+/+	−/−	−/−	+/+
P2	−/−	−/−	+/+	−/−	−/−	+/+
P3	−/−	−/−	+/+	−/−	−/−	+/+
P4	+/−	−/−	+/+	−/−	−/−	+/+
P5	+/−	−/−	+/+	−/−	−/−	+/+
P6	+/−	−/−	+/+	−/−	−/−	+/+
P7	−/−	−/−	+/+	−/−	−/−	+/+
P8	−/−	−/−	+/+	−/−	−/−	+/+
P9	+/−	−/−	+/+	−/−	−/−	+/+
P10	−/−	−/−	+/+	−/−	−/−	+/+
P11	+/−	−/−	+/+	−/−	−/−	+/+
C1	−/−	−/−	+/+	−/−	−/−	−/−
C2	+/+	−/−	+/+	−/−	−/−	−/−
C3	+/−	−/−	+/+	−/−	−/−	−/−

(−/−) absence of variant; (+/−) heterozygous for variant; (+/+) homozygous for variant; (P) RP/CRD patient; (C) control.

**Table 3 ijms-25-01004-t003:** Patients and Diagnoses.

Patient	Alternate ID †	Gender	Diagnosis **
P1	CHRD5308	M	RP
P2	CHRD4047	M	RP
P3	CHRD3323	M	RP
P4	CHRD0262	M	RP
P5	*	F	RP
P6	CHRD5151	M	RP
P7	CHRD0677	M	RP
P8	CHRD3458	M	RP
P9	MOL0884-2	F	CRD
P10	MOL0884-1	F	CRD
P11	MOL0884-4	M	CRD
C1	*	F	control
C2	*	F	control
C3	*	F	control

† See [13] * No previous alternate ID in literature. ** RP (rod-cone dystrophy); CRD (cone-rod dystrophy).

**Table 4 ijms-25-01004-t004:** Primers used for potential modifier gene PCR amplification.

Gene	Forward Primer	Reverse Primer	Size (nt)
*ALG6*	5′-TCTAGTAGCTTCCTGCTCCCT	5′-ATCCTTTGGAAGAGGGCTGAA	575
*ALG8*	5′-GCTGTCTTTCAGAGATGATGCAA	5′-GCCACCCAAACATAAAGGAGC	198
*DDOST*	5′-GTGGCCGATCCTGATAACCC	5′-CCAGCAATGAGGAGGGTGTT	373
*MPDU1*	5′-CTGCTTCCTGGTCATGCACT	5′-GGGTGACTACAGTCAAGGGC	240
*TNKS*	5′-TTGTGTGGCTTCCCTAGGTTTG	5′-CTTCACAGTTTCCAAGTCTCCA	276
*DHDDS*	5′-TCACCTTGGGGTGTAGTGTCT	5′-AACACTCTCCAACCACAGCAA	291

## Data Availability

All data presented here are available from the corresponding author (SJP) upon justifiable request. Relevant patient data is similarly available with appropriate de-identification.

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
