# Peer review of "Inherited Retinal Degeneration Caused by Dehydrodolichyl Diphosphate Synthase Mutation–Effect of an ALG6 Modifier Variant"

_ijms, 2024, doi:10.3390/ijms25021004_

Round 1

Reviewer 1 Report

Comments and Suggestions for Authors

I will congratulate the authors to an extremely important report  regarding  modifying gene in patients with DHDDS. You demonstrate that  modifier  genes may play a role in severity of disease expression and perhaps in future to solve the problem about syndromic and non-syndromic patient with the same genotype.

According your report

·       Identifying the same mutation in the same  genotype, with different modifying genes!

·       Examining the phenotype in similar ways during several years (foveal photoreceptor thickness, visual acuity, foveal sensitivity, macular and extramacular rod sensitivity and kinetic visual field extent)  The type of equipment changes during the years , but authors  try in best way follow the phenotype during long period of time

·       Even if the  author not give answer  to the prognosis  and difference in phenotypes it point out the importance of modifier genes

Some questions

The phenotype in these disorders are mostly described as rod cone degeneration, but I miss some convincing data about that. The phenotypical  examinations of  structure and function in the  macular  and pericentral area was clearly described , but what about periphery. It was mention about ERG and visual field in periphery. If possible some of data (figure) could be added to verify the total retinal function and of  cone rod/rod cone degeneration  

Before the genotype in these group of patients were identified and especially  syndromic patients, you could measure CDT Carbo hydrate-deficient serum transferrin (asialic or disialicform). Perhaps was hat the normal and of no value in this group of patients?

Author Response

Reviewer 1

  1. “The phenotype in these disorders are mostly described as rod cone degeneration, but I miss some convincing data about that. The phenotypical examinations of structure and function in the macular and pericentral area was clearly described , but what about periphery. It was mention about ERG and visual field in periphery. If possible some of data (figure) could be added to verify the total retinal function and of cone rod/rod cone degeneration”

Response: We appreciate the Reviewer’s comment. We have now revised the results to include these results in 10 of 11 patients, both rod and cone ERGs were severely attenuated, and one patient showed a cone-worse-than-rod CRD phenotype on ERG. The extent of peripheral visual field is shown in Fig.3E. We hope this is satisfactory.

Reviewer 2 Report

Comments and Suggestions for Authors

In general, the presented manuscript is relevant and is devoted to a rare but severe group of diseases. Authors describe sequencing results of 29 DNA samples from 11 patients and the results of quantitative measures of retinal structure and function which were performed across five decades of life. The study design is consistent with the research questions. The statistical analysis was carried out correctly. The figures and tables accurately reflect the results of the study.

During the review, some questions arose:

1.      In chapter 2.3. “Differences in severity of disease in patients” two patients underwent a complete examination necessary for the described inherited retinal degeneration, the results of the examination were described and illustrated in detail. The age of these patients is indicated (24 years), but the age of manifestation of the disease is not indicated.

2.      The study did not describe the results of the ERG. The authors explain this by saying that the study results were limited, but did not explain a reason why the results were limited.

This article does not contain fundamental comments, has a novelty in the topic under study and can be recommended for publication in open sources.

Author Response

  1. “In chapter 2.3. “Differences in severity of disease in patients” two patients underwent a complete examination necessary for the described inherited retinal degeneration, the results of the examination were described and illustrated in detail. The age of these patients is indicated (24 years), but the age of manifestation of the disease is not indicated.

Response: Based on this comment, we have now revised this section to include the age of onset of symptoms reported by the patients.

  1. “The study did not describe the results of the ERG. The authors explain this by saying that the study results were limited, but did not explain a reason why the results were limited.”

Response: As mentioned above in response to Reviewer 1 comments, we have now revised the results to include more ERG information. Specifically, in 10 of 11 patients, both rod and cone ERGs were severely attenuated, and one patient showed a cone-worse-than-rod dystrophy (CRD) phenotype on ERG.

  1. “Before the genotype in these group of patients were identified and especially syndromic patients, you could measure CDT Carbo hydrate-deficient serum transferrin (asialic or disialicform). Perhaps was that the normal and of no value in this group of patients?”

Response: We did not consider the mechanism of disease in this manuscript and thus did not include this analysis. Furthermore, as the reviewer indicates, no changes in glycosylation were observed in other studies reporting on DHDDS patients with retina or brain anomalies and in our analysis of a DHDDS K42E knockin mouse model (Lam, Zuchner et al., 2014, Adv. Exp. Med. Biol. 801:165-170.; Kim, Kim et al., 2021, Ann. Clin. Transl., Neurol. 8:2319-2326; Hamdan et al., 2017, Am. J. Hum. Genet. 101:664-685; and our K42E knockin mouse model, Ramachandra Rao et al., 2020, Cells 9:896-906).